# Antibody Evaluation and Mutations of Antigenic Epitopes in the Spike Protein of the Porcine Epidemic Diarrhea Virus from Pig Farms with Repeated Intentional Exposure (Feedback)

**DOI:** 10.3390/v14030551

**Published:** 2022-03-07

**Authors:** Thu Hien Nguyen Thi, Chi-Chih Chen, Wen-Bin Chung, Hso-Chi Chaung, Yen-Li Huang, Li-Ting Cheng, Guan-Ming Ke

**Affiliations:** 1International Degree Program of Animal Vaccine Technology, International College, National Pingtung University of Science and Technology, No.1, Shuefu Road, Neipu, Pingtung 91201, Taiwan; thuhien120185@gmail.com; 2Research Centre for Animal Biologics, National Pingtung University of Science and Technology, No. 1 Shuefu Road, Neipu, Pingtung 91201, Taiwan; chen0100@gmail.com (C.-C.C.); wbchung@mail.npust.edu.tw (W.-B.C.); hcchaung@mail.npust.edu.tw (H.-C.C.); ellis7374365@yahoo.com.tw (Y.-L.H.); 3Graduate Institute of Animal Vaccine Technology, College of Veterinary Medicine, National Pingtung University of Science and Technology, No. 1 Shuefu Road, Neipu, Pingtung 91201, Taiwan

**Keywords:** porcine epidemic diarrhea, phylogenetic analysis, lactogenic immunity, spike gene, antigen epitopes, feedback

## Abstract

The feedback strategy, or controlled exposure of pig herd to the porcine epidemic diarrhea virus (PEDV), significantly decreased losses during a severe outbreak in late 2013 in Taiwan. However, some pig farms still suffered from recurrent outbreaks. To evaluate the association between antibody titers and clinical manifestations, sera and colostra were analyzed from one pig farm that employed the feedback strategy. Furthermore, spike (S) gene full sequences from six positive samples of two farms with and without using feedback were compared to investigate the evolution of PEDV variants circulating in pig herds. The results in this study showed that high PEDV antibody titers do not correlate with the high rate of protection from PEDV infection. In addition, repeated feedback generated the emergence of PEDV variants with unique substitutions of N537S and Y561H in the COE domain and S769F in the SS6 epitopes. These mutations indicated the pathogenetic evolution of PEDV strains existing in the cycle of the feedback method. A very strict biosecurity practice to block the routes of pathogen transfer should be followed to achieve successful control of PEDV infections in pig herds.

## 1. Introduction

Porcine epidemic diarrhea (PED) is a highly transmissible disease that causes substantial economic losses to pork producers all over the world [1,2,3]. PED clinical signs include watery diarrhea, vomiting, dehydration, high morbidity, and mortality in the suckling piglets. The causative agent, porcine epidemic diarrhea virus (PEDV), is an enveloped, single-stranded, positive-sense RNA virus, belonging to the family *Coronaviridae*, subfamily *Coronavirinae*, genus *Alphacoronavirus*. The genome of PEDV is approximately 28 kb in length and comprises 2 overlapping open reading frames (ORFs) 1a/1b encoding for 16 non-structural proteins (nsps), then a series of downstream ORFs encoding the 4 structural proteins, spike (S), envelope (E), membrane (M), and nucleocapsid (N), plus an accessory protein gene ORF3 [4]. The PEDV S protein is a homotrimeric membrane glycoprotein that is composed of two subunits, S1 and S2, mediating receptor binding and cell membrane fusion, respectively [5,6]. The essential roles of S protein in host–virus interactions indicate that these proteins may contain antigenic epitopes capable of inducing functional protective immunity against PEDV infection in pigs. Previous reports have shown that the S protein contains four antigenic regions: a CO equivalent (COE) domain, and epitopes SS2, SS6, and 2C10. Studies on the molecular mechanisms of evolution, pathogenicity, and antigenicity of PEDV variants circulating in pig populations focus, therefore, upon these antigenic regions [6,7,8,9].

PED was first recorded in the UK in 1971, and the virus was identified in Belgium in 1978 [10]. The new PEDV variants quickly spread to many Asian countries and later to the USA [11,12]. Indeed, one report estimated that approximately seven million piglets died from PEDV infection during the 2013 to 2014 outbreaks in the USA [13].

A severe outbreak of PED occurred in late 2013 in Taiwan. Genetic analysis showed that the PEDV strains isolated were closely related to those causing severe outbreaks in the USA in early 2013 [14]. Due to a lack of an effective vaccine, a “feedback” strategy, or controlled exposure of all sows and gilts to PEDV, was conducted in numerous farms to control the disease. The feedback strategy was expected to stimulate maternal PEDV, which would be passively transferred to piglets through colostrum/milk, thus protecting vulnerable piglets from PEDV infection [15,16,17]. The feedback strategy significantly reduced losses during the acute outbreaks and resulted in the elimination of PEDV from many pig herds. However, some pig farms—especially those with continuous flow management systems—suffered from ongoing clinical diseases with recurrent outbreaks. This raised the question of whether the lactogenic immunity induced by feedback provided sufficient protection of piglets from PEDV infection. The first subject to be addressed in this study was, therefore, to analyze the relationship between passively transferred PEDV antibody titers and the occurrence of clinical disease in piglets.

The feedback strategy can be helpful for the induction of herd immunity against PEDV infection, but the strategy also means that the pig herds are exposed to an evolving population of viruses. This suggests that antigenic variation of PEDV in response to immense immune pressure might frequently arise and could quickly lead to the generation of PEDV variants. The second subject of this study aims to investigate the evolution and emergence of PEDV variants in pig herds under repeated virus exposure during feedback feeding of PEDV. The information provided by this study will aid in the development of strategies for vaccine development and the control of PEDV.

## 2. Materials and Methods

### 2.1. Pig Farms and Sample Collection

Three commercial farrow-to-finish pig farms, SL, RS, and CL, each with 1500 sows, located in southern Taiwan, were selected in this study. All three farms had severe outbreaks of PED during early January 2014. The study was approved by the Institutional Animal Care and Use committee of the National Pingtung University of Science and Technology. (IACUC-NPUST-106-059).

Blood was collected from pigs via jugular venipuncture using vacutainer tubes. After collection, each serum sample was divided into three parts of 0.5 mL and stored at −20 °C. Colostrum was sampled by an experienced pig handler manually from sows within 18 h post-partum. Approximately 15 mL of sample was collected and divided into three 5 mL centrifuge tubes and stored at −20 °C for IgA analyses. Intestinal samples, including feces, were collected from acutely infected piglets with clinical signs including anorexia, vomiting, and severe watery diarrhea. The samples were homogenized, diluted 1:5 in phosphate-buffered saline (PBS), and centrifuged at 10,000× *g* at 4 °C for 5 min. Supernatants were divided into aliquots of 1 mL each and stored at −80 °C.

### 2.2. Cells and Virus

Vero cells (ATCC number: CCL-81) (USA) were cultured in DMEM (Dulbecco’s modified eagle medium, Life Technologies, Carlsbad, CA, USA) supplemented with 10% fetal bovine serum (FBS), penicillin (100 U/mL), and streptomycin (100 U/mL) at 37 °C in a 5% CO_2_ incubator. 

From June to December 2020, intestinal tissues and feces were collected from piglets with severe watery diarrhea at SL and CL pig farms. Samples were collected in a sterile bag and submitted to the Research Center for Animal Biologics (RCAB), National Pingtung University of Science and Technology, Taiwan, for further processing and analyses. All samples were positive for PEDV by reverse transcription-polymerase chain reaction (RT-PCR). PEDV strains TW/PT01/2020 (PT01), TW/PT02/2020 (PT02), TW/PT03/2020 (PT03), TW/PT04/2020 (PT04), and TW/PT05/2020 (PT05) were obtained from SL farm and TW/CY4/2020 (CY4) from CL farm (Table 1).

### 2.3. Serology

The concentrations of IgG in serum or IgA in colostra were detected by using a commercial PEDV ELISA kit (GENTAUR, Kampenhout, Belgium) and AniGen PED IgA ELISA (BioNote Inc., Hwaseong-si, South Korea), respectively, and performed following the manufacturer’s instructions. 

The neutralizing antibody titers of PEDV in sera and colostra were evaluated following a previous study [18], with some modifications. The sera and milk of the sows were diluted in two-fold serial dilution in serum-free DMEM medium before mixing with an equal volume of 200 TCID_50_ PEDV and incubated for 1 h at 37 °C, then transferred to 96-well plates containing confluent Vero cell monolayers. After 2 days, CPE was observed using an inverted microscope. 

### 2.4. RNA Extraction and RT-PCR

Samples were identified as PEDV-positive by the conventional RT-PCR method, which targets the ORF3 gene, and confirmed again by quantitative reverse transcription-polymerase chain reaction (RT-qPCR), performed as previously described [9,19].

From the positive samples, RNA of the virus was extracted by the AxyPrep Body Fluid Viral DNA/RNA Miniprep Kit (Bioscience, Unioncity, CA, USA). The cDNA was immediately synthesized with random primers by the High-Capacity cDNA Reverse Transcription Kit (ThermoFisher, Carlsbad, CA, USA). 

### 2.5. Sequencing of S Gene and Phylogenetic Analysis

Sequences encoding the S gene (4161 bp) were amplified by PCR using GDP-HiFi DNA polymerase (GeneDireX, Las Vegas, NV, USA), with specific primers designed based on the sequences of the PEDV reference (Table 2), following a previous study [20] under the following conditions of adjusted thermal cycles: initial denaturation at 95 °C for 5 min, 35 cycles of denaturation at 95 °C for 30 s, annealing at 60 °C for 30 s, extension at 72 °C for 4 min 30 s, and a final extension at 72 °C for 5 min. The amplified PCR products were purified by 1% agarose gel electrophoresis. The PCR product was excised from the gel then purified using FavorPrep GEL/PCR Extraction (Favorgen Biotech Corporation, Pingtung, Taiwan). The purified PCR product was then ligated into the pGEM T-Easy vector system (Promega, Fichburg, WI, USA). The identity of clones was confirmed by automated DNA sequencing (Genomics, New Taipei City, Taiwan).

Multiple sequence alignments were performed using ClustalW (www.ebi.a-c.uk/clustalw accessed on 27 December 2021) with the Lasergene MegAlign (DNASTAR, Madison, WI, USA) and MEGA software programs. Phylogenetic trees were constructed using the molecular evolutionary genetics analysis MegAlign version 6.0.

### 2.6. Data Analysis and Statistics

All data of the SN titers in serum were log-transformed as the base of 2 to show potential linear associations. Data are expressed as mean ± SEM. A Student’s *t*-test was performed to compare treatment means between groups, and the serum antibody titers of each sow before and after the feedback were analyzed by the paired *t*-test. The serum antibody levels in piglets were categorized at three levels: low (L: SN ≤ 8 or ELISA IgG ≤ 25, IgA ≤ 1.5), medium (M: 8 < SN ≤ 32, 25 < ELISA IgG ≤ 40, 1.5 < ELISA IgA ≤ 2.5), and high (H: SN ≥ 32, or ELISA IgG ≥ 40, IgA ≥ 2.5). The associations of the categorical variables with the subsequent incidence of diarrhea were analyzed using the χ^2^ test. Differences in which *p* < 0.05 were considered statistically significant.

## 3. Results

### 3.1. Lactogenic Immunity Increased after Intentional Exposure (Feedback)

To investigate the immune responses of sows after intentional exposure to PEDV, sera were collected from 15 sows before and after feedback in farm SL. Results showed that a significant (*p* < 0.05) increase in antibody titers was observed either examined by the SN test (3.7 ± 0.6 and 21.5 ± 5.8 for before and after feedback) or ELISA (20.1 ± 2.7 and 27.5 ± 5.0 for before and after feedback) (Table 3). Sera were also collected from two offspring of each sow seven days after farrowing. The SN titers against PEDV of piglet serum (77.2 ± 16.5) were significantly (*p* < 0.05) higher than those of sow serum (21.5 ± 5.8), and with a moderate positive correlation (r = 0.42) between each other.

### 3.2. Lactogenic Immunity Showed No Association with the Protection of Piglets from PEDV in the Fields

To determine the associations between antibody titers and clinical manifestations, the antibody titer distributions in pigs with and without diarrhea were compared. Sera were collected from 2 piglets 2 days after birth from each of 16 and 15 sows in farms SL and RS, respectively. Colostra were also collected from 85 sows on the day of farrowing in the SL farm. Piglets with clinical signs of diarrhea caused by PEDV were confirmed by RT-PCR. Results showed that there was no association of the serum antibody levels with diarrhea in piglets either determined by SN (χ^2^ *p*-values of 0.15 and 0.78 for farms SL and RS, respectively) or ELISA IgG (χ^2^ *p*-values of 0.30 and 0.39 for farms SL and RS, respectively) (Figure 1). There were also no associations of the IgA antibody titers in sow colostra with the protection of piglets from PEDV infection observed with χ^2^ *p*-values of 0.40 for farm SL (Figure 2).

### 3.3. Phylogenetic and Genetic Analysis

The full-length S gene of PEDV strains collected from SL and CL farms was analyzed to determine their phylogenetic and genetic relationship with 34 sequences of PEDV strains from GenBank (Table A1). The phylogenetic tree was divided into two major groups (G1 and G2) (Figure 3). Phylogenetic analysis shows that the six strains in this study cluster within the G2b group, closely related to the Japanese strain (IBR-7/JPN/2014), but less related to other Asian strains, such as Chinese PEDV strains (CH/GSTS/2016), Vietnamese PEDV strains (HUA-PED45), and Korean PEDV strains (CNUP1F-2019). However, they differ genetically from European strains (including CV777). The six strains analyzed in this study form a different sub-group from the previous virus strains isolated in Taiwan between 2013–2014 and 2016–2018.

The full length of the S gene of all five strains from farm SL is 4161 bp (1386 aa). The nucleotide sequences are separated into two closely related lineages: Group A (PT01, 02, 05) and Group B (PT 03, 04). When aligning to the CV777 vaccine strain, the identification of five strains had several point mutations, showing a low nucleotide sequence identity at 93.39% to 93.63%. 

In comparison with the previous Taiwan PED strains circulating between 2013–2014 and 2016–2018, the five strains collected in our study shared between 99.15% to 99.61% nucleotide sequence identities.

**Figure 3 viruses-14-00551-f003:**
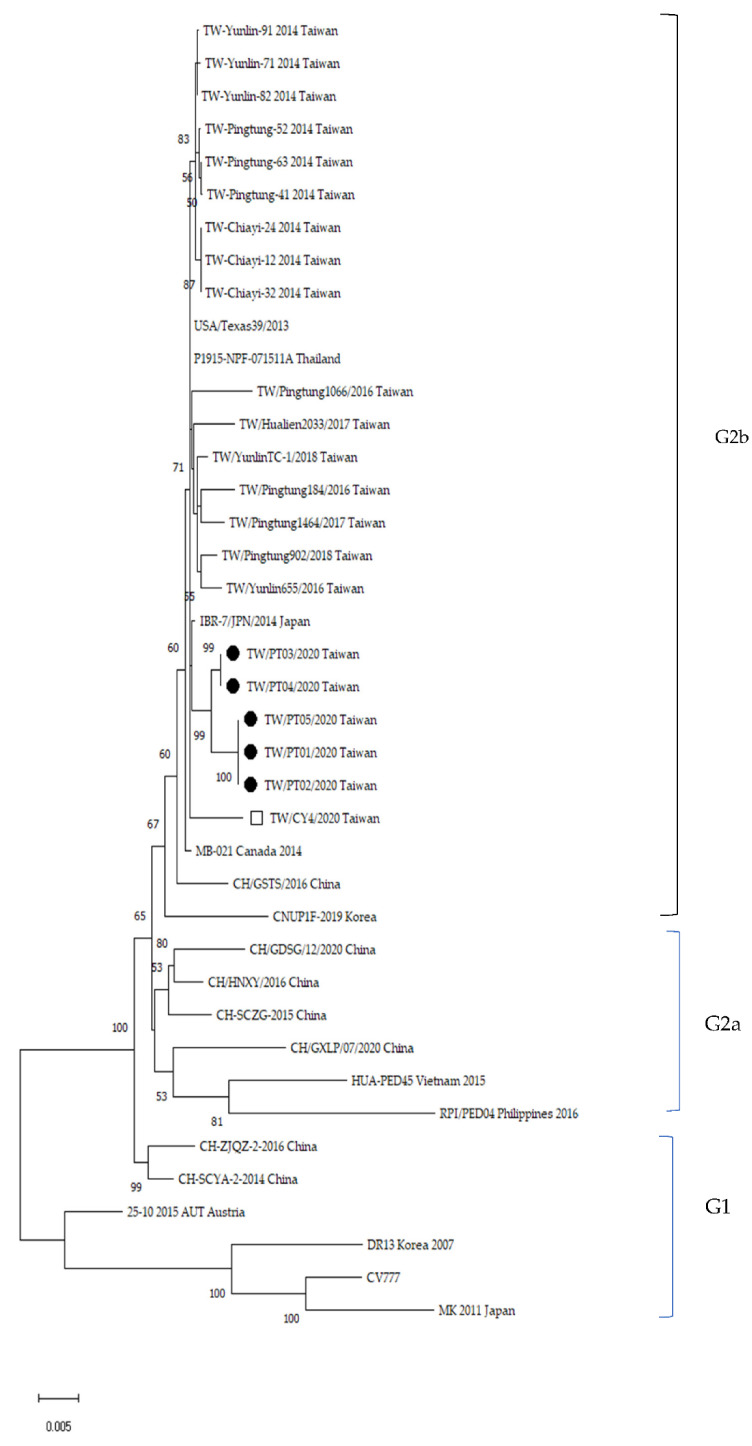
Phylogenetic tree of PEDV strains based on full-length spike gene generated by the Maximum Likelihood method with the Kimura two-parameter model, using the MEGA software. Bootstrapping with 1000 replicates was performed to determine the percentage reliability for each internal node. Horizontal branch lengths were correlated to genetic distances among PEDV strains, with different genogroups labeled on the right. Black solid circles represent PEDV strains isolated from the SL farm that used feedback. The hollow square represents CY4 strains isolated from the CL farm with no feedback. The scale bar represents nucleotide substitutions per site.

### 3.4. Alignment of Antigenic Epitopes of S Protein

The S protein comprises a total of 1386 amino acids for all six PEDV strains obtained from SL and CL farms. In this study, we focused on analyzing the antigenic regions containing epitopes capable of inducing neutralizing antibodies against PEDV. The amino acid sequence corresponding to the COE domain and epitopes SS2, SS6, and 2C10, were located in the residues 502–641, 751–758, 767–774, and 1371–1377, respectively [21,22,23]. 

Corresponding to the phylogenetic analysis shown in Figure 3, the five PEDV strains collected from the SL farm can be categorized into two groups when analyzed using protein sequence alignment software. Complete identities in the S protein were observed amongst the PT01, PT02, and PT05 strains (Group A) and between the PT03 and PT04 strains (Group B). A total of 13 substitutions (P27S, A169D, A174T, S182H, M214T, L299F, S769F, P885S, A1054V, P1094L, R1104K, G1225S, and E1382K) in the S protein were observed when comparing the amino acid sequences of Group B PEDV strains with those of Group A. The two groups share 99.06% sequence identity. The Group A and Group B strains exhibit 98.2% and 98.7% identity with CY4, respectively.

Overall, the sequence alignment of the S proteins demonstrated that the Taiwanese topotype strain TW-Pingtung-41 (GenBank no. KP276251.1) and the six strains identified in this study share 92.22% to 93.01% of amino acid identity with the PEDV prototype CV777. Both CY4 and Group B strains exhibit 98.85% of amino acid identity with Taiwanese topotype strain TW-Pingtung-41, but Group A strain shares a slightly lower identity of 98.34%. 

Multiple alignments of amino acid sequences of each antigenic region in S proteins among the identified field PEDV strains in this study and those of 35 Taiwanese strains published in GenBank were conducted (Table 4). The amino acid sequence of the COE domain of CY4 shares 100% identity with that of strain TW-Pingtung-41, while both Group A and Group B strains share only 98.57%. As compared with CY4 and 35 Taiwanese PEDV strains, the 5 strains from SL farms were uniquely characterized by substitution mutations of N537S and Y561H in the COE domain. Amino acid variations at residues D509A, H524P/S, K566N, L603I, Y606H, E608D/A, G612V/A/D, and E636V in the COE region were occasionally identified among those of published Taiwanese PEDV strains. Another unique substitution (S769F) in epitope motif SS6 of Group B strains but not of Group A was also observed. No mutations were identified in epitopes SS2 and 2C10.

## 4. Discussion

Intentional exposure of sows and gilts to PEDV via feedback of homogenized intestines could offer protective maternal immunity to the most susceptible suckling piglets and thus shorten the outbreak period and minimize economic losses on pig farms. However, determined by the method of feedback, the farm types and sizes, the herd management systems, and biosecurity measures implemented, the intentional virus-exposure practices might result in a negative or positive impact on pig farms [24,25]. Therefore, our research at periodically used feedback farms aimed to: (i) evaluate the relationship between passive humoral immunity and the time showing clinical signs and (ii) investigate the evolution of the PEDV variants that circulated in this farm.

The lactogenic immunity to protect neonatal suckling piglets via colostrum and milk is critical in the prevention and control of PEDV infection [26]. Several studies support that exposure to infected PEDV tissues can induce a significant increase in specific IgA and IgG levels [17,18,19]. Our data from the SL farm are consistent with these reports. Results from the SN and ELISA IgA analyses in this study showed significantly higher antibody levels in sow sera after feedback. In the SL farm, sows were given oral exposure to PEDV-infected materials. This practice might provide mucosal immunization. The previous studies showed that boosting inactivated virus in previously exposed sows may increase protection in nursing piglets against PEDV and maternally derived PEDV-neutralizing activity. The specific memory B cells could develop subsequent intentional exposure [27,28]. Memory B cells might be critically related to the ability to induce and secrete mucosal-specific antibodies [29,30]. In our study, PEDV-specific antibodies (IgA and IgG) were detected in colostrum and milk, implying that memory B cells responsible for IgA and IgG synthesis existed at the intestinal mucosa and can respond to this PEDV stimulation. Colostrum/milk antibody levels associated with a protective immune response in piglets may be a more reliable diagnostic tool than systemic antibody responses for detecting vaccination efficacy [27,31]. 

In piglets, humoral immunity is important for protection against PED. In one recent study at a Midwestern United States sow farm, the sows were given oral immunization by feedback after an initial PEDV outbreak. Two months after this feedback approach, all weaned pigs in that farm tested negative for PEDV [32]. It has been shown that the immune response of pregnant sows vaccinated with the PEDV S1 protein provided lactogenic immunity to the piglets through colostrum and milk [33]. However, the correlation between the antibody levels and the time of showing clinical signs in piglets is not clearly described. Few researchers have considered IgA antibody responses as a correlate of protection [28,34,35]. The previous report has also demonstrated that anti-PEDV antibodies through intraperitoneal administration could contribute to the protection of piglets against PEDV infection and the passively transferred circulating antibodies partially improved the course of infection [36]. On the other hand, in our study, there was no association of the serum antibody levels with the diarrhea symptom either determined by the SN test or ELISA IgG. Our results also indicated no associations of the IgA antibody titers in sow colostrum with the protection of piglets from PEDV infection. Moderate or even severe PED occurred in piglets with high titers of maternal antibody in the SL pig farm, indicating that the viral loads in the barn environment might exceed the protection threshold provided by the passively transferred lactogenic immunity. Current PEDV intentional exposure is frequently ineffective, and it should be noted that: (i) there is no standard, optimized, feedback protocol, (ii) the potential of contamination of the feedback material with other infectious pathogens, (iii) mutations altering the antigenic nature of feedback viruses, and (iv) the possibility of continuous re-infection. Indeed, several studies in recent years have indicated that hazard to the herd arises from the re-infecting virus (using feedback or live vaccines) from reversion or recombination to virulence [17,37,38]. Therefore, the feedback protocols should only be applied in urgent explosive outbreaks to initiate and maintain a high lactogenic immunity. Disinfection and strict biosecurity measures should be thoroughly followed for a successful control and eradication of PEDV in pig herds. 

PEDV phylogenetic investigation in Taiwan recently confirmed that the primary PEDV strain is related to Genogroup 2b, closely related to the strains USA/Texas39/2013 and Thailand P1915-NPF-01511A [14,20]. Although the strains isolated in this study are closely related with the US Genogroup 2b, they have greater similarity with the Japanese strain IBR7/JPN/2014 than with Chinese or earlier Taiwanese strains. Currently, very limited research has been conducted on the evolution of the PEDV circulating on the farm using feedback. While there have been multiple reports on the incidence and genetic diversity of the PED virus in Taiwan, no studies of the virus circulation in closed farms that used feedback regularly have been conducted. This research displayed several point mutations in the full spike gene of five virus strains—notably potential significant mutations of antigenic epitopes in the S gene. From our study, PEDV strains sequenced in that farm in 2020 have several mutations within the S gene compared to earlier Taiwanese strains (2013–2014 and 2016–2018). Phylogenetic analysis categorized the five PEDV strains identified from the SL farm into two groups, showing that two variants coexist in the pig population. The coexistence of different PEDV variants within pig herds or even individual pigs with high frequency has been reported [18,39]. The repeated feedback practices implemented in the SL farm might promote the emergence of new PEDV variants. The CY4 strain identified from the CL farm without frequent feedback exhibited 100% identity on the COE domain of S protein as compared with the topotype strain TW-Pingtung-41, but the five strains identified from the SL farm shared only 98.57%. These results revealed the fact that a complicated, constantly mutated, mixed genotype infection of PEDV might occur in pig herds utilizing a repeated feedback program.

The antigenic epitopes inducing neutralizing antibodies against PEDV infection in pigs are mainly located in the S glycoprotein. The mutations therefore more frequently occur in the S protein of PEDV [40,41,42]. Mutations in the S protein can lead to a variety of structural and functional consequences and contribute to the lack of efficacy of PEDV vaccines [42,43,44,45]. In this study, the two particular substitutions, N537S and Y561H, within the COE domain of the S protein could be significant in the pathogenetic nature of the PEDV strain isolated from the SL farm. It is interesting to note that lethal strains identified in China, South Korea, Japan, Taiwan, Canada, Mexico, and the United States all have common substitutions (-LQDGQVKI- to -SQSGQVKI-) within the SS6 epitope region of the S1 gene, which may play a role in PEDV virulence [19]. However, in our study, the unique substitution S→F (-SQSGQVKI- to -SQFGQVKI-) on the SS6 region of Group B strains could be an important mutation point to analyze the B cell epitopes on the S gene. Interestingly, no such above-mentioned mutations were found in PEDV strain CY4 identified from the CL farm without frequent feedback, and only 10 km distant from the SL farm during the same period. Furthermore, the SL farm was a tightly closed herd, and the replacement breeding stock came from only one source farm free from PEDV infection. The PEDV strains with substitution mutations would be unlikely to originate from other pig herds. The above information suggests that the amino acids at the positions of 537, 561, and 769 in the antigenic regions of S protein might be the most susceptible residues subject to substitution mutation in pig herds under immense immune pressure of the host and the high viral loads in the environments. Further examination of the molecular structure of these mutations should be carried out to understand the significance of such changes within the antigenic epitopes of PEDV in more detail. Similar to another report [46], the sequences of SS2 (aa 751–758) and 2C10 (aa 1371–1377) motifs were quite conserved, and no substitution in these motifs was found in this study. 

## 5. Conclusions

In the case of PEDV infection, passively transferred maternal antibodies cannot provide full protection against the virus in piglets under field conditions with high viral loads. A very strict biosecurity program should be followed to achieve successful control of PEDV infections in pig herds with feedback practice. PEDV variants with unique substitution mutations of N537S and Y561H in the COE domain and S769F in SS6 epitopes emerged in pig herds under repeated intentional exposure of PEDV to sows and gilts.

## Figures and Tables

**Figure 1 viruses-14-00551-f001:**
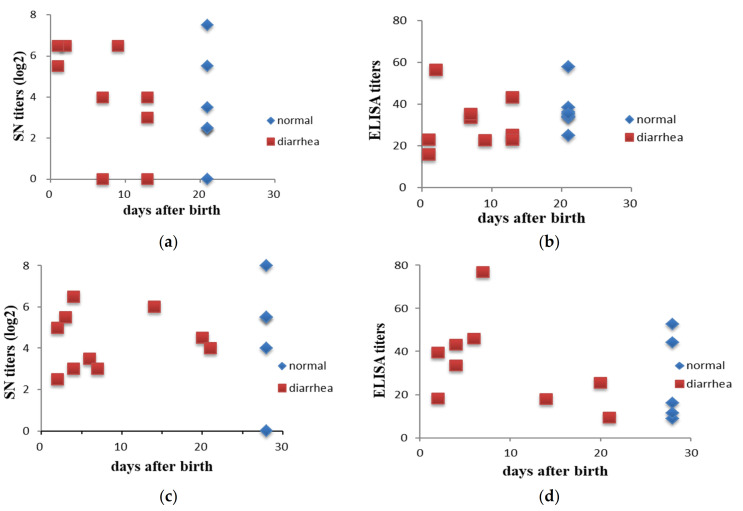
Comparison of the distribution of serum neutralization and ELISA antibody titers in piglets with or without diarrhea in farms SL (**a**,**b**) and RS (**c**,**d**). Each point represents one litter and sera were collected from 2 piglets 2 days after birth from each sow. For the SL farm, *n* = 9 and 7 for litters with or without diarrhea, respectively, and for the RS farm, *n* = 10 and 5 for litters with or without diarrhea, respectively.

**Figure 2 viruses-14-00551-f002:**
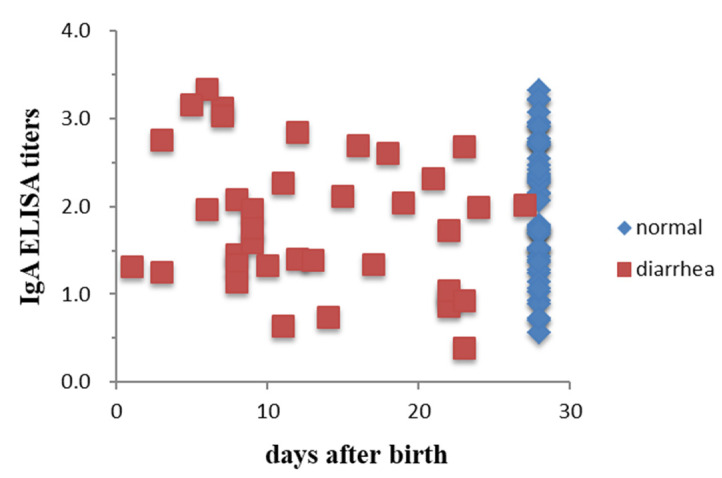
Comparison of the distribution of ELISA IgA antibody titers against PEDV in piglets with or without diarrhea in farm SL. Each point represents one litter, and colostrum was collected from sows on the day of farrowing (*n* = 37 and 48 for litters with or without diarrhea, respectively).

**Table 1 viruses-14-00551-t001:** Field strains of PED virus collected from piglets with diarrhea.

Virus Strain	Abbreviation	Pig Farm	Age of Pig (Days)	Sample	Isolated Time
TW/PT01/2020	PT01	SL	4	Intestine	August 2020
TW/PT02/2020	PT02	SL	>7	Intestine	October 2020
TW/PT03/2020	PT03	SL	5	Intestine	September 2020
TW/PT04/2020	PT04	SL	6	Intestine	December 2020
TW/PT05/2020	PT05	SL	3	Feces	December 2020
TW/CY4/2020	CY4	CL	21	Intestine	December 2020

**Table 2 viruses-14-00551-t002:** List of primers used in the study.

Primer Name	Nucleotide Sequence, (5′-3′)	Targets
Full S (F)	TAAGTTGCTAGTGCGTAATAATGAC	S full-length amplification
Full S (R)	CAGACTTCGAGACATCTTTG	
PED.sq1	CTTTGTTAGCCATATTAGAGGT	Sequencing of the spike gene
PED.sq2	ACGATCGATGGTGTTTGTAATGGA	
PED.sq3	GGACACTAATTGCCCTTTCACCT	
PED.sq4	ACGCCTGTTAGTGTTGATTG	
PED.sq5	TTGAGAGTGTTAAAGAGGCTATT	
PED.sq6	ACTCTCGACTGGACATTC	
PED.sq7	AGCCATTTCTAGTTCTATTG	

**Table 3 viruses-14-00551-t003:** Serum antibody titers against PEDV before and after feedback of sows.

Detection Methods	No. of Sows	Antibody Titers	
		Before Feedback	2 Weeks after Feedback
Serum neutralization	15	3.7 ± 0.6 ^a,^*	21.5 ± 5.8 ^b^
ELISA	15	20.1 ± 2.7 ^a^	27.5 ± 5.0 ^b^

* Data are presented as mean ± SEM. ^a,b^ Means with different superscript letters in a row are significantly different (*p* < 0.05).

**Table 4 viruses-14-00551-t004:** Alignment summary of amino acid sequences of the antigenic regions in S protein among the strains identified in this study and those of Taiwanese PEDV strains published in GenBank.

GenBank No. or Strain Name	Year First Identified	Substitution Sites in Antigenic Regions of S Protein *																							
		COE Domain (Residues 502–641)																							SS6 (Residues 767–774)
		509	520	524	526	531	532	537	541	554	561	566	573	581	594	603	604	606	608	612	615	624	626	636	769
KP276251.1 (15) **	2013	D	S	H	G	A	S	N	S	F	Y	K	P	D	L	L	F	Y	E	G	F	K	E	E	S
KJ434295.1	2013	•	•	P	•	•	•	•	•	•	•	•	•	•	•	I	•	•	D	•	•	•	•	•	
KM246672.1	2013	•	•	P	•	•	•	S	•	•	•	•	•	•	•	I	•	•	D	•	•	•	•	•	
KM246707.1	2013	•	•	•	S	•	•	•	•	•	•	•	•	N	•	•	•	•	•	•	•	•	•	•	
KP276252.1	2014	•	•	•	•	•	•	•	•	•	•	N	•	•	•	•	•	•	•	•	•	•	•	•	
KJ434308.1	2014	•	•	•	•	•	•	•	•	•	•	•	L	•	•	•	•	•	•	•	•	•	•	•	
KJ434306.1	2014	•	•	•	•	•	•	•	•	•	•	•	•	•	•	•	•	•	•	•	•	E	•	•	
KY929405.1	2015	A	•	•	•	•	•	•	•	•	•	N	•	•	•	•	•	•	•	•	•	•	•	•	
KY929406.1	2016	A	•	•	•	•	•	•	•	S	•	N	•	•	•	•	•	•	•	•	•	•	•	•	
MK673523.1	2016	•	•	•	•	•	•	•	•	•	•	•	•	•	•	•	•	•	•	•	•	•	•	V	
MK673526.1	2017	•	•	•	•	•	•	•	•	•	•	N	•	•	•	•	L	•	•	A	•	•	•	•	
MK673531.1	2017	•	•	•	•	•	•	•	•	•	•	N	•	•	•	•	•	•	•	V	•	•	•	•	
MK673517.1	2017	•	•	•	•	•	•	•	•	•	•	•	•	•	•	•	•	•	•	•	•	•	•	V	
MK673535.1	2017	•	•	P	•	•	I	•	Y	•	•	•	•	•	•	•	•	H	D	•	•	•	•	•	
MK673545.1	2018	•	•	•	•	V	•	•	•	•	•	•	•	•	•	•	•	•	•	•	•	•	•	•	
MK673528.1	2018	•	•	•	•	•	•	•	•	•	•	•	•	•	•	•	•	•	D	V	•	•	•	•	
MK673527.1	2018	•	•	•	•	•	•	•	•	•	•	•	•	•	•	•	•	•	D	V	•	•	•	•	
MK673520.1	2018	•	•	•	•	•	•	•	•	•	•	N	•	•	•	•	•	•	•	•	•	•	•	•	
KM673538.1 (2)	2018	•	•	•	•	•	•	•	•	•	•	N	•	•	F	•	•	H	A	D	•	•	•	•	
KM246732.1	2018	•	F	S	•	•	•	•	•	•	•	•	•	•	•	•	•	•	D	•	Y	•	D	•	
CY4	2020	•	•	•	•	•	•	•	•	•	•	•	•	•	•	•	•	•	•	•	•	•	•	•	
PT01 (3)	2020	•	•	•	•	•	•	S	•	•	H	•	•	•	•	•	•	•	•	•	•	•	•	•	
PT03 (2)	2020							S			H														F

* Amino acid sequence alignment was conducted using NCBI BLASTp (blast.ncbi.nlm.nih.gov accessed on 27 December 2021) with KP276251.1 serving as the reference strain. Only those residues with differences are shown in the table. ** The number in the parentheses indicates the number of strains with complete identities.

## Data Availability

The data that support the findings of this study are available in the main manuscript of this article.

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
