# Peer review of "Antibody Evaluation and Mutations of Antigenic Epitopes in the Spike Protein of the Porcine Epidemic Diarrhea Virus from Pig Farms with Repeated Intentional Exposure (Feedback)"

_viruses, 2022, doi:10.3390/v14030551_

Round 1
Reviewer 1 Report
In this MS, the authors evaluated the association between PEDV antibody titers and clinical manifestations in pig farm employed the feedback strategy and the pathogenetic evolution of PEDV strains existing in the cycle of feedback method. The results provided some valuable information for understanding the feedback strategy for control PEDV infection. But the authors need to improve their article to make it acceptable for publication.
- The title of the MS should be “Antibody Evaluation and Mutations of Antigenic Epitopes in the Spike Protein of the 2 Porcine Epidemic Diarrhea Virus from Pig Farms with Repeated Intentional Exposure (Feedback)”.
- The isolated time should be provided of the six field strains of PEDV in this study.
- The names of virus family, subfamily and genus should be italicized.
Author Response
Response to Reviewer # 1 comments
Comments and Suggestions for Authors
In this MS, the authors evaluated the association between PEDV antibody titers and clinical manifestations in pig farms employed the feedback strategy and the pathogenetic evolution of PEDV strains existing in the cycle of feedback method. The results provided some valuable information for understanding the feedback strategy for control PEDV infection. But the authors need to improve their article to make it acceptable for publication.
- The title of the MS should be “Antibody Evaluation and Mutations of Antigenic Epitopes in the Spike Protein of the 2 Porcine Epidemic Diarrhea Virus from Pig Farms with Repeated Intentional Exposure (Feedback)”.
Response 1: The title has been edited as “Antibody Evaluation and Mutations of Antigenic Epitopes in the Spike Protein of the Porcine Epidemic Diarrhea Virus from Pig Farms with Repeated Intentional Exposure (Feedback)”.
- The isolated time should be provided for the six field strains of PEDV in this study.
Response 2: The isolated time has been added in Table 1. (page 3)
- The names of virus family, subfamily and genus should be italicized.
Response 3: The names of virus family, subfamily, and genus have been italicized. (page 1, line 35-36)
Reviewer 2 Report
Piglets infected with PEDV will produce watery diarrhea with high mortality. The feedback of sows in contact with PEDV is a means to prevent and control PED in some pig farms. This paper evaluated the relationship between the antibody level of sows and the infection of piglets with PEDV. Some mutations in the important epitope of PEDV spike protein in the pig farms that employed feedback strategy were found, which is important for the design of PEDV vaccine in the future. But there are some issues that need to be clarified.
1. The types of PEDV strains and their spike protein important epitopes used in pig farms(SL、RS and CL).
2. In results about the relationship between pig antibody titer and diarrhea in Figure 1, there are data from SL and RS pig farms, but there is no PEDV strain from RS pig farm in Table 1.
3. TW/PT04/2020 (CY4) from CL farm (Table 1) in line 100 should be TW/CY4/2020 (CY4) from CL farm (Table 1).
4. Is Table 3 it in line 154? Is not Table 2 it in 154?
5. Is RS (c and d), not RS (c and b), it in line 175?
6. The location of the epitope described in line 215-217 should be referenced.
Author Response
Response to Reviewer # 2 comments
- The types of PEDV strains and their spike protein important epitopes used in pig farms(SL, RS, and CL).
Response 1: In this study, we did not sequence any PEDV strain from the RS farm. Only antibody analysis was performed on samples from RS farm. Therefore, we only provide strain information and epitope analysis for strains from SL and CL farms. The types and antigenic epitopes in the spike protein of the PEDV strains identified from SL and CL farms were described in the manuscript:
- Genetic analysis of the spike protein of the PEDV strains collected in Taiwan during the years 2013 and 2014 showed that the strains from SL and CL farms are closely related to those strains identified in the US in early 2013 (page 10, line 309). The outbreaks of PED in these three pig farms (SL, RS, and CL) occurred during the same period of early January 2014 (page 2, line 80). Therefore, the PEDV strains circulated in farms SL, RS and CL are most likely originated from the same topotype similar to the strain TW-Pingtung-41 (Table 3).
- The phylogenetic analysis shows that the six strains identified in this study are related to Genogroup 2b (page 6, line 189).
- The amino acid sequence corresponding to the COE domain, epitopes SS2, SS6, and 2C10, were located in the residues 502–641, 751–758, 767–774, and 1371–1377, respectively (page 8, line 217).
- In results about the relationship between pig antibody titer and diarrhea in Figure 1, there are data from SL and RS pig farms, but there is no PEDV strain from RS pig farm in Table 1.
Response 2: Since we did not sequence any PEDV strain from the RS farm, there is no RS farm data in Table 1. Only antibody analysis was performed on samples from RS farm. CL farm is located only 10 km away from SL farm in the same pig raising area (page 11, line 341) however, RS farm is located in another pig raising area far away from SL farm. We use the strain (CY4) identified from the CL farm to demonstrate the unique PEDV strain with specific mutations in spike gene was not of a geographically endemic strain but a strain specifically emerged from the pig herd under repeated feedback (page 11, line 347).
- TW/PT04/2020 (CY4) from CL farm (Table 1) in line 100 should be TW/CY4/2020 (CY4) from CL farm (Table 1).
Response 3: The error was corrected (page 3, line 101).
- Is Table 3 in line 154? Is not Table 2 it in 154?
Response 4: The error was corrected (page 4, line 155).
- Is RS (c and d), not RS (c and b), it in line 175?
Response 5: The error was corrected (page 5, line 176).
- The location of the epitope described in line 215-217 should be referenced.
Response 6: The citations [21-23] of the location of the epitopes have been added (page 8, line 217)